# Mask What Matters: Controllable Text-Guided Masking for Self-Supervised Medical Image Analysis

## Abstract

The scarcity of annotated data in specialized domains such as medical imaging presents significant challenges to training robust vision models. While self-supervised masked image modeling (MIM) offers a promising solution, existing approaches largely rely on random high-ratio masking, leading to inefficiency and poor semantic alignment. Moreover, region-aware variants typically depend on reconstruction heuristics or supervised signals, limiting their adaptability across tasks and modalities. We propose Mask What Matters, a controllable text-guided masking framework for self-supervised medical image analysis. By leveraging vision-language models for prompt-based region localization, our method flexibly applies differentiated masking to emphasize diagnostically relevant regions while reducing redundancy in background areas. This controllable design enables better semantic alignment, improved representation learning, and stronger cross-task generalizability. Comprehensive evaluation across multiple medical imaging modalities, including brain MRI, chest CT, and lung X-ray, shows that Mask What Matters consistently outperforms existing MIM methods (e.g., SparK), achieving gains of up to +3.1 percentage points in classification accuracy, +1.3 in box average precision (BoxAP), and +1.1 in mask average precision (MaskAP) for detection. Notably, it achieves these improvements with substantially lower overall masking ratios (e.g., 40% vs. 70%). This work demonstrates that controllable, text-driven masking can enable semantically aligned self-supervised learning, advancing the development of robust vision models for medical image analysis.

## 1 INTRODUCTION

Image classification, segmentation, and object detection are fundamental tasks in computer vision. In recent years, machine learning—especially deep learning—has become the core technology driving advances in this field. However, high-performing deep learning models typically rely on large-scale annotated datasets. In specialized domains such as medical image processing, high-quality annotations are often scarce and expensive to obtain, posing a significant barrier to the widespread adoption of deep learning methods.

Self-supervised learning (SSL) offers a promising solution by using unlabeled data for pretraining, enabling models to learn effective feature representations without manual annotations Jaiswal et al. (2020); Liu et al. (2021); Krishnan et al. (2022); Huang et al. (2023). Among various SSL paradigms, MIM has emerged as a popular approach. By randomly masking parts of an image and requiring the model to reconstruct the original content, MIM encourages the model to capture structural and semantic patterns from visible context. It has demonstrated strong performance across a range of downstream tasks and has been successfully extended to medical image analysis He et al. (2022); Gupta et al. (2025); Xiao et al. (2023).

However, prevailing MIM approaches, such as masked autoencoders (MAE) He et al. (2022) and SparK Tian et al. (2023)—typically adopt random masking strategies, which lack alignment with regions of interest in downstream tasks. This task-agnostic masking introduces two main limitations: (1) The learned representations may be semantically misaligned with the downstream objectives, reducing transfer performance. (2) To increase the chance of masking informative areas,

existing methods often use extremely high masking ratios (e.g., 75% in MAE), which increases reconstruction difficulty and demands large-scale data and computational resources. In contrast, natural language exhibits compact structure and high semantic density, allowing effective learning even at low masking ratios Devlin et al. (2019). The reliance on high masking ratios in image-based MIM largely stems from inherent information redundancy and the lack of semantic-aware masking, which makes it difficult for models to focus on truly critical regions.

This issue is particularly pronounced in medical imaging. Medical images are characterized by high semantic sparsity, where diagnostically relevant information is often confined to small localized regions (e.g., lesions or organs), while a large portion of the image consists of semantically redundant background. Additionally, medical imaging spans diverse modalities—including MRI, CT, and X-ray—with substantial variation in the shape, size, and spatial distribution of task-relevant areas across different tasks. These properties impose stricter requirements on the efficiency and generalizability of masking strategies.

To address these challenges, we introduce a controllable, text-guided masking framework—hereafter referred to as **Mask-What-Matters (MWM)**. Given a user-specified, open-vocabulary description or phrase, a pretrained vision–language model provides zero-shot localization cues that we convert into robust regions of interest (ROIs). MWM then applies region-specific, prompt-conditioned masking—assigning higher ratios to semantically important areas (e.g., lesions/organs) and lower ratios to background—thereby injecting downstream semantics directly into the pretraining signal without per-image reports or labels.

To summarize, our main contributions are:

- The first controllable text-guided masking framework for medical imaging. MWM integrates vision–language models (BiomedCLIP) with segmentation refinement (SAM) to localize task-relevant regions from open-vocabulary prompts and apply differentiated ROI vs. background masking, overcoming the semantic misalignment of random masking.

- An annotation-free and backbone-agnostic design. MWM requires no per-image reports or labels, and can be seamlessly integrated into both ViT- and ConvNet-based MIM pipelines under a unified protocol.

- Consistent gains across modalities and tasks. On brain MRI, chest CT, and X-ray datasets, MWM surpasses state-of-the-art MIM baselines (e.g., SparK), improving classification, detection, and segmentation performance and demonstrating the promise of text-driven generative pretraining in medical imaging.

## 2 RELATED WORK

### 2.1 MASKED IMAGE MODELING

Masked modeling originated in the natural language processing domain with the bidirectional encoder representations from transformers (BERT) model Devlin et al. (2019), which learns contextual representations by masking tokens and predicting them during pretraining. Inspired by this idea, He et al. (2022) introduced MIM to computer vision, where portions of an image are masked during pretraining and the model is tasked with reconstructing the missing regions. This paradigm has demonstrated strong generalization in both the natural and medical image domains (Gupta et al., 2025; Liu et al., 2021; Huang et al., 2023). Representative methods include MAE (He et al., 2022), MedMAE (Gupta et al., 2025; Xiao et al., 2023), and SparK (Tian et al., 2023). Among them, MAE utilizes high-ratio random masking to encourage ViTs to focus on high-level semantic representations; MedMAE extends this idea to medical images with domain-specific adaptations; while SparK adapts the strategy to convolutional backbones, leveraging their inherent multi-scale features to boost pretraining effectiveness.

Despite these advances, conventional MIM approaches typically adopt random or structure-agnostic masking strategies that ignore the semantic requirements of downstream tasks. Such non-selective masking may lead the model to overemphasize irrelevant regions during pre-training, reducing transfer effectiveness. To address this issue, recent work has explored region-aware masking strategies aimed at improving semantic alignment. Two main directions have emerged:

One line of work introduces loss-driven adaptive masking, where methods such as hard patches mining (HPM) Wang et al. (2023a) and its extension AnatoMask Li et al. (2024) dynamically analyze reconstruction errors across regions and assign higher masking probabilities to areas with larger losses, aiming to prioritize semantically valuable structures. However, this approach can mislead the model toward hard-to-reconstruct but task-irrelevant regions. For example, in brain magnetic resonance imaging (MRI), complex structures like the spinal cord or eyeballs may attract unnecessary attention despite being unrelated to downstream tasks like tumor detection.

Another line of work incorporates external perceptual modules to guide masking. For example, FocusMAE Basu et al. (2024) leverages pretrained Region Proposal Networks (RPNs) to identify high-information regions for selective masking. While effective in certain tasks, this strategy requires supervised training of RPNs on domain-specific annotated data, limiting its cross-task generalization.

These limitations highlight the need for a more flexible, semantically aligned and generalizable masking mechanism. Natural language, as a rich and interpretable source of supervision, presents a promising direction for guiding masked modeling in a task-aware yet annotation-free manner.

## 2.2 VISION-LANGUAGE PRE-TRAINED MODELS

Benefiting from the rapid development of vision-language alignment models in recent years, it has become increasingly feasible to accurately perceive and identify task-relevant regions under zero-shot and open-world settings Radford et al. (2021). Classic approaches such as the Contrastive Language-Image Pretraining (CLIP) family achieve a unified embedding space for images and texts through large-scale joint pretraining on image–text pairs, and have been widely adopted in zero-shot and open-set visual tasks. In the medical imaging domain, extensions like BiomedCLIP Zhang et al. (2023) further enhance cross-modal understanding, achieving robust performance across diverse downstream applications. Although vision-language pretrained models have been widely adopted in various supervised learning tasks, most existing studies still focus on downstream applications such as object detection and semantic segmentation (Zhong et al., 2022; Aleem et al., 2024; Koleilat et al., 2024), without exploring how their open-vocabulary and zero-shot capabilities could benefit masked image modeling.

To summarize, although recent region-aware MIM approaches—such as dynamic masking based on reconstruction loss or masking guided by external perception modules—have improved semantic focus to some extent, they still suffer from several limitations: reliance on additional supervision, misalignment between masked regions and task semantics, and lack of flexibility. Meanwhile, despite the strong performance of vision-language models in downstream tasks, little effort has been made to incorporate natural language prompts into the MIM pretraining phase for task-aware masking guidance. Consequently, these observations point to a promising direction—incorporating vision-language semantic guidance to enable efficient and generalizable masked image modeling.

## 3 METHODOLOGY

### 3.1 OVERALL FRAMEWORK

As illustrated in Figure 1, **Mask What Matters (MWM)** comprises two core stages: (1) text-guided region localization and (2) region-aware masked image modeling. The first stage leverages a frozen vision–language model to identify task-relevant regions from an open-vocabulary prompt, while the second stage applies differentiated masking ratios across regions to guide the reconstruction process.

### 3.2 TEXT-GUIDED REGION LOCALIZATION

MWM localizes task-relevant regions through a multi-stage pipeline. It begins with LLM-based prompt generation and employs BiomedCLIP to extract cross-modal embeddings. A saliency map is then produced via the Multi-Modal Information Bottleneck (M2IB), binarized with K-Means, and refined using the Segment Anything Model (SAM). Finally, the refined mask is converted into a bounding box with a controllable expansion margin, which serves as a robust spatial guide for the subsequent masked modeling stage.

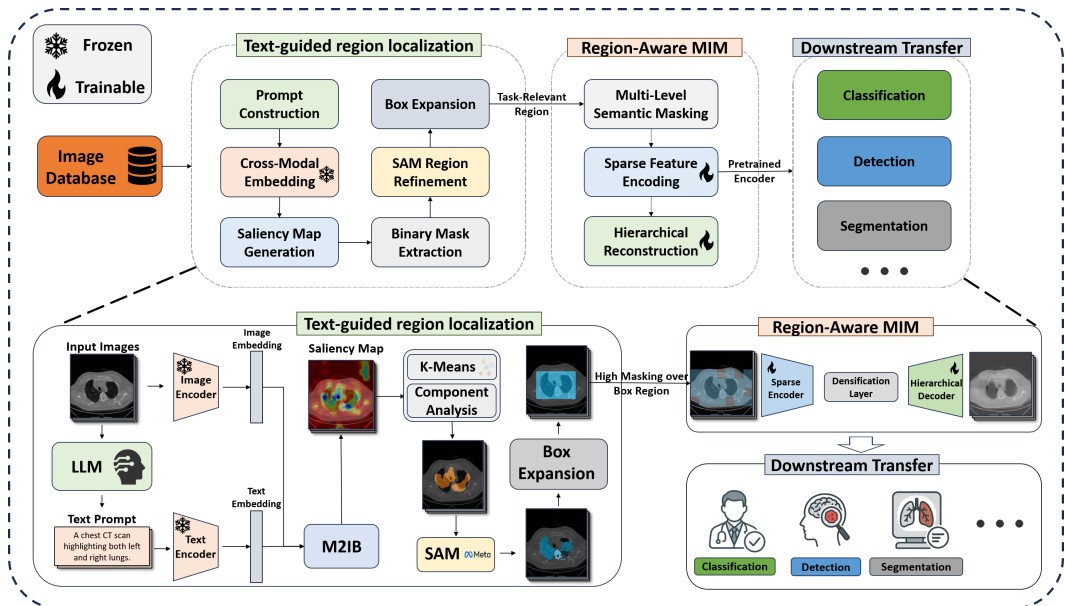

Figure 1: Overview of the MWM framework. The top panel shows the three-stage pipeline: (1) text-guided region localization, (2) region-aware masked image modeling, and (3) downstream transfer. The bottom panel zooms in to illustrate how prompts guide region localization and masking.

**Prompt Generation.** In medical image analysis, the design of textual prompts plays a critical role in determining the effectiveness of region localization. To enhance semantic guidance, large language models (LLMs) are used with structured prompts such as "Describe the typical visual characteristics of a [*category*] in a [*modality*] image" is queried. In addition to these full-sentence prompts, simple category phrases are also experimentally employed as alternative inputs. A detailed comparison of these prompt styles is presented in the experimental section.

**Cross-Modal Embedding.** After the prompt is generated, a BiomedCLIP model Zhang et al. (2023) is employed to encode the medical image and its prompt into a shared semantic space. Given an image $I$ and a prompt $T$, the embeddings are computed as:

$$Z_{\text{img}} = \Phi_{\text{img}}(I), \quad Z_{\text{text}} = \Phi_{\text{text}}(T) \tag{1}$$

where $\Phi_{\text{img}}$ and $\Phi_{\text{text}}$ denote the vision and language encoders, respectively. These cross-modal embeddings provide the semantic foundation for saliency estimation in subsequent stages.

**Saliency Map Generation.** A cross-modal saliency estimation module based on the Multi-modal Information Bottleneck (M2IB) Wang et al. (2023b) is exploited to localize task-relevant visual regions by aligning image and text embeddings while suppressing modality-specific redundancy. Formally, the saliency map $\lambda_S \in [0,1]^{H \times W}$ assigns importance weights to each spatial location based on its semantic relevance to the prompt. M2IB achieves this by optimizing the following objective:

$$\mathcal{L}_{\text{M2IB}} = \text{MI}(Z_{\text{img}}, Z_{\text{text}}) - \gamma \cdot \text{MI}(Z_{\text{img}}, I) \tag{2}$$

where $Z_{\text{img}}$ and $Z_{\text{text}}$ are image and text embeddings encoded by BiomedCLIP respectively, and $\gamma$ controls the trade-off between preserving cross-modal relevance and filtering out task-irrelevant visual information.

**Binary Mask Extraction.** After obtaining the saliency map, we binarize it using unsupervised K-Means clustering Lloyd (1982) to localize the region guided by the text prompt, generating a preliminary foreground mask. Specifically, the map is clustered into two pixel groups, and the cluster with higher saliency values is identified as foreground. To further refine the mask and suppress false activations, connected component analysis is performed on the binary map, retaining only the largest foreground regions. This step helps filter out noisy edge areas and focuses attention on the core semantic structures.

**Region Refinement.** To enhance spatial accuracy, SAM is incorporated as a refinement module Kirillov et al. (2023). For each selected connected region $c_i \in \mathcal{C}^*$, its minimal enclosing bounding box is computed to serve as the visual prompt for SAM.

Given the original image $I$ and the set of bounding boxes, SAM predicts a refined segmentation mask:

$$M_{\text{SAM}} = \text{SAM}(I, \text{Box}(c_i)) \tag{3}$$

The resulting mask $M_{\text{SAM}}$ provides more faithful spatial structure.

**Box-Based Region Expansion.** The SAM mask obtained from the previous stage contains boundary noise, which can mislead the model's focus during representation learning. To improve the robustness of region guidance, the refined SAM mask is converted into a bounding box with a controllable expansion margin, which serves as the final region of interest (ROI) for downstream masked modeling.

### 3.3 Region-Aware Masked Image Modeling

Given the region of interest generated by the localization module in Section 3.2, MWM performs masked image modeling through a three-stage pipeline: (1) a multi-level masking strategy that applies differentiated masking ratios based on region importance, (2) a sparse encoder that processes only the unmasked patches, and (3) a hierarchical decoder that reconstructs the full image from sparse multi-scale features.

**Multi-Level Semantic Masking.** MWM adopts a multi-level masking strategy that leverages the semantic regions identified by the text-driven localization process. Specifically, patches corresponding to regions highlighted by the textual prompts—such as regions associated with tumors or organs—are masked at a high ratio to encourage the model to focus on reconstructing task-relevant semantic features. In contrast, background regions are masked lightly or left unmasked to reduce redundant computation. A visual comparison of masking patterns is shown in Figure 2.

**Sparse Feature Encoding.** Visible patches are collected into a sparse image and encoded following the SparK design Tian et al. (2023):

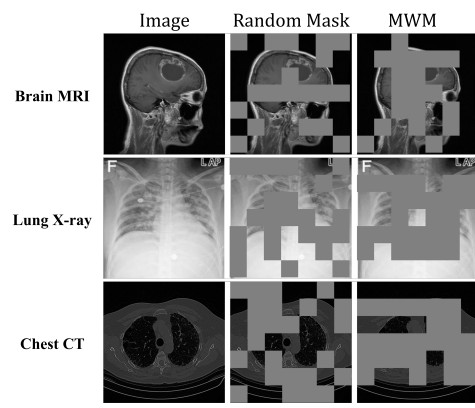

Figure 2: Comparison of masking strategies. Gray blocks indicate masked regions.

$$X_{\text{sparse}} = \{x_i \mid m_i = 1\},$$
$$F = \Phi_{\text{sparse}}(X_{\text{sparse}}) \tag{4}$$

where $x_i$ is the $i$-th patch, $m_i \in \{0, 1\}$ indicates visibility, $\Phi_{\text{sparse}}$ is the encoder, and $F = \{F_1, F_2, F_3, F_4\}$ are multi-scale features. While we instantiate MWM with convolutional backbones, the same interface applies to ViTs via token indexing.

**Hierarchical Reconstruction.** A lightweight UNet-style decoder reconstructs the full image in a top-down manner. At each scale $l$, masked locations are filled with a learned embedding $M_l$:

$$F_l'(i) = \begin{cases} F_l(i), & \text{if patch } i \text{ is visible,} \\ M_l, & \text{if patch } i \text{ is masked.} \end{cases} \tag{5}$$

The densified map $F_l'$ is projected by a scale-specific layer $\phi_l$ to obtain $D_l$, then upsampled and fused with features at the next lower level:

$$D_{l-1} = B_{l-1}(D_l) + \phi_{l-1}(F_{l-1}'). \tag{6}$$

Finally, the model is trained with mean-squared error on masked patches:

$$\mathcal{L}_{\text{recon}} = \frac{1}{|\mathcal{I}_{\text{mask}}|} \sum_{i \in \mathcal{I}_{\text{mask}}} \|\hat{x}_i - x_i\|^2, \tag{7}$$

where $\mathcal{I}_{\text{mask}}$ is the set of masked patch indices, $x_i$ the original patch, and $\hat{x}_i$ its reconstruction.

# 4 EXPERIMENTS

## 4.1 DATASETS

**Self-Supervised Pretraining Datasets.** Three datasets spanning different imaging modalities are used for self-supervised pretraining. (1) *Brain MRI* includes approximately 17,000 images from Brain Tumor MRI Cheng et al. (2015), BRISC Fateh et al. (2025), and BraTS 2018 Menze et al. (2014), covering various tumor types and anatomical structures. (2) *Lung X-ray* consists of around 39,000 chest radiographs from COVID-QU-Ex Rahman et al. (2021); Chowdhury et al. (2020), including COVID-19, non-COVID pneumonia, and normal cases. (3) *Chest CT* comprises 12,000 slices from a lung disease dataset Konya (2020), with segmentation masks of fibrotic lungs from 107 patients.

**Downstream Fine-Tuning Datasets.** Three datasets are used for downstream evaluation across classification and detection tasks. (1) *Brain Tumor MRI* Feltrin (2023) includes 4,479 MRI images spanning 44 tumor types, such as astrocytoma, glioblastoma, meningioma, and ependymoma. (2) *Pediatric Lung X-ray* Kermany et al. (2018) consists of 5,863 pediatric chest X-rays categorized as either normal or pneumonia. (3) *Chest CT Cancer* Hany (2020) includes CT images from four classes: adenocarcinoma, large cell carcinoma, squamous cell carcinoma, and normal tissue.

## 4.2 EXPERIMENTAL SETUP

All self-supervised methods use backbones of similar model scale, with convolution-based methods adopting ResNet-50 He et al. (2016) and transformer-based methods using ViT-S Dosovitskiy et al. (2020). The input resolution is 224×224, and models are initialized with ImageNet-pretrained weights. Pretraining uses an initial learning rate of $2\times10^{-4}$. Downstream fine-tuning retains the same encoder and input resolution. All experiments are performed on NVIDIA Tesla A100 cards with 80 GB VRAM under Python 3.9, Ubuntu 22.04.3, and PyTorch 1.10.0. Unless otherwise specified, we fix non-target variables (e.g., prompt type or SAM usage) to their empirically optimal settings during evaluation to ensure controlled comparison.

## 4.3 EVALUATION OF TEXT-GUIDED LOCALIZATION

We first evaluate the effectiveness of the text-guided localization module across three imaging modalities—brain MRI, lung X-ray, and chest CT—and further examine the impact of prompt type and SAM refinement on localization performance. Specifically, we compare the predicted bounding boxes with expert-annotated segmentation masks, and report occlusion precision ($|R_p \cap R_{gt}|/|R_p|$) and occlusion recall ($|R_p \cap R_{gt}|/|R_{gt}|$), where $R_p$ is the predicted region and $R_{gt}$ is the ground-truth annotation.

As shown in Table 1, our text-guided localization with effective prompts achieves consistently high occlusion recall across all datasets (around 0.82), while maintaining reasonable precision. This

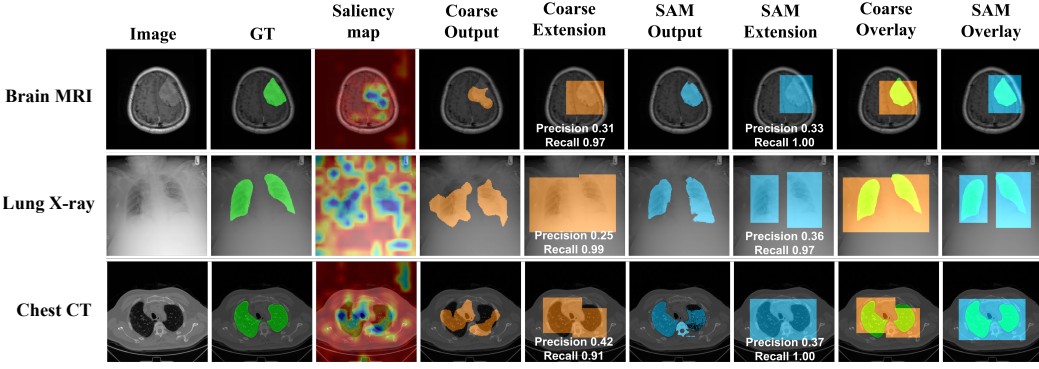

Figure 3: Representative results of text-guided region localization.

| Method | Brain MRI | | Chest CT | | Lung X-ray | |
|---|---|---|---|---|---|---|
| | Prec. | Rec. | Prec. | Rec. | Prec. | Rec. |
| *Prompt Type* | | | | | | |
| Phrase | 0.22 | 0.86 | 0.21 | **0.99** | 0.42 | **0.82** |
| Descrip. Sent. | **0.30** | **0.94** | **0.37** | 0.88 | **0.44** | 0.66 |
| *Impact of SAM* | | | | | | |
| Coarse Ext. | 0.22 | **0.95** | 0.36 | 0.87 | 0.37 | **0.84** |
| + SAM Ext. | **0.30** | 0.94 | **0.37** | **0.88** | **0.42** | 0.82 |

Table 1: Localization performance (Occlusion Precision/Recall) across prompt types and refinement strategies. *Coarse Ext.* denotes bounding box extension of the coarse region; + *SAM Ext.* indicates further refinement using SAM.

confirms the framework's ability to identify task-relevant regions solely from textual descriptions, despite modality-specific anatomical variations (see Figure 3 for visualized examples).

**Effect of Prompt Type on Region Localization.** To further assess the impact of language design, we evaluate the effect of two prompt types on localization performance. As shown in Table 1, descriptive sentence prompts yield better results in Brain MRI and Chest CT, indicating that richer semantics aid localization. In contrast, concise phrases outperform in lung radiographs in terms of recall (0.82 vs. 0.66), likely because the lungs' broad anatomical coverage allows general terms to be sufficiently effective.

**Effect of SAM Refinement on Region Localizatio.** We also evaluate the contribution of spatial refinement by applying SAM to the coarse masks. As shown in Table 1, SAM refinement improves precision while maintaining high recall. For instance, in Brain MRI, precision increases from 0.22 to 0.30, while recall remains above 0.94. This demonstrates the utility of SAM in refining coarse localization outputs.

## 4.4 COMPARISON WITH PREVIOUS SSL METHODS

We then evaluate the generalization and representation ability of MWM by comparing it with representative self-supervised learning (SSL) methods across both reconstruction-based and contrastive learning paradigms. All methods are pretrained and evaluated using consistent datasets and training settings to ensure a fair comparison.

**Image Classification.** Classification performance on three downstream medical imaging datasets is examined. Following common practice, results are reported under two settings: (1) *full fine-tuning*, where the entire model is updated; and (2) *linear probing*, where the encoder is frozen and only a linear classifier is trained on top.

- **Full fine-tuning:** Table 2 summarizes classification accuracy for MWM and several representative self-supervised methods, including MAE He et al. (2022), SparK Tian et al. (2023),

| Method | Type | Brain MRI | Chest CT | Lung X-ray |
|---|---|---|---|---|
| MoCoV2 | CL | 95.3 | 91.7 | 93.1 |
| BYOL | CL | 91.3 | 93.7 | 93.9 |
| SimCLR | CL | 94.5 | 93.1 | 94.1 |
| MAE | MIM | 95.4 | 91.7 | 94.9 |
| AnatoMask | MIM | 96.5 | 95.8 | 95.2 |
| SparK | MIM | 96.2 | 94.4 | 94.7 |
| **MWM(Ours)** | MIM | **96.8** | **97.5** | **96.0** |

Table 2: Fine-tuning classification accuracy (%) on downstream datasets, where MIM stands for Masked Image Modeling, while CL stands for Contrastive Learning.

| Method | Brain MRI | Chest CT | Lung X-ray |
|---|---|---|---|
| AnatoMask | 70.2 | 63.9 | 85.3 |
| SparK | 69.8 | 63.5 | 81.6 |
| **MWM(Ours)** | **71.3** | **67.0** | **88.5** |

Table 3: Linear probing accuracy (%) on downstream datasets (frozen encoder).

| Method | Detection | | Segmentation | |
|---|---|---|---|---|
| | $AP^{box}$ | $AP^{box}_{75}$ | $AP^{mask}$ | $AP^{mask}_{75}$ |
| AnatoMask | 45.2 | 53.1 | **46.2** | 54.1 |
| SparK | 45.6 | 53.2 | 44.8 | 52.0 |
| **MWM (Ours)** | **46.9** | **53.5** | 45.9 | **54.1** |

Table 4: Detection ($AP^{box}$, $AP^{box}_{75}$) and instance segmentation ($AP^{mask}$, $AP^{mask}_{75}$) results (%) on the BR35H dataset.

AnatoMask Li et al. (2024), SimCLR Chen et al. (2020a), MoCoV2 Chen et al. (2020b), and BYOL Grill et al. (2020). MWM consistently achieves the highest performance, demonstrating strong generalization across diverse imaging modalities. The most significant gain is observed on the Chest CT dataset, where MWM achieves 97.5% accuracy—surpassing SparK by +3.1 and AnatoMask by +1.7. This improvement highlights the benefit of text-guided masking, which emphasizes semantically critical yet spatially sparse regions—such as lung lobes in CT scans—while avoiding masking redundancy.

- **Linear probing:** To further evaluate the quality of learned representations, linear probing experiments are conducted. For simplicity, MWM is compared with two recent state-of-the-art masked image modeling methods—SparK and AnatoMask. As shown in Table 3, MWM significantly outperforms both across all datasets, highlighting its ability to learn effective features through text-driven pretraining.

**Object Detection and Instance Segmentation.** To assess the generalizability of learned representations beyond classification, we evaluate brain tumor detection and instance segmentation on the BR35H dataset Hamada (2025). As shown in Table 4, **MWM** achieves the highest $AP^{box}$ (46.9%) and the second-highest $AP^{mask}$ (45.9%), surpassing SparK by +1.3 and +1.1 points, respectively. These results demonstrate that MWM transfers effectively to both object-level localization and pixel-wise segmentation.

In summary, incorporating text-driven masking during pretraining enables the model to learn more informative and transferable representations. This semantic guidance consistently improves performance across classification, detection, and segmentation tasks, underscoring its effectiveness in enhancing visual representation learning.

### 4.5 ABLATION STUDY

We also conduct ablation experiments to evaluate the individual impact of prompt design, SAM-based region refinement, and masking ratio on downstream classification performance in MWM.

#### 4.5.1 EFFECT OF PROMPT DESIGN AND REGION REFINEMENT

As shown in Table 5, descriptive sentence prompts outperform short phrases on brain MRI and chest CT, while the opposite trend is observed on lung X-ray. This aligns with the localization results in Section 4.3, where prompts that enabled more accurate region identification also led to better downstream results. Importantly, both types of prompts—regardless of granularity—consistently outperform random masking across all datasets. This confirms that text-guided masking enables effective semantic alignment between pretraining and downstream tasks, which is crucial for improving self-supervised learning performance. In addition, an ablation on SAM demonstrates that incorporating this module during pretraining consistently improves downstream classification, confirming its effectiveness in enhancing text-guided masking.

| Group | Setting | Brain MRI | Chest CT | Lung X-ray |
|-------|---------|-----------|----------|------------|
| Prompt Type | No Prompt | 96.2 | 94.4 | 94.7 |
| | Phrase | 96.4 | 96.5 | **96.0** |
| | Sentence | **96.8** | **97.5** | 95.2 |
| Impact of SAM | w/o SAM | 96.4 | 96.8 | 95.8 |
| | w/ SAM | **96.8** | **97.5** | **96.0** |

Table 5: Downstream classification accuracy (%) under different prompt types and region refinement strategies across three imaging modalities. *No Prompt* refers to random masking without text-guided localization. *w/* and *w/o* denote with and without SAM, respectively.

### 4.5.2 EFFECT OF MASKING RATIO

As shown in Figure 4, MWM achieves 96.8% classification accuracy at a masking ratio of just 40%—a setting substantially lower than around 70% typically used in masked image modeling. Despite the reduced ratio, it outperforms SparK's best performance at 70% masking by +2.4% in absolute accuracy. To rule out the possibility that SparK's inferior performance is due to suboptimal masking configurations, SparK under 40%, 50%, 60%, and 70% masking ratios are further evaluated. The results confirm that 70% masking yields the highest accuracy for SparK, validating

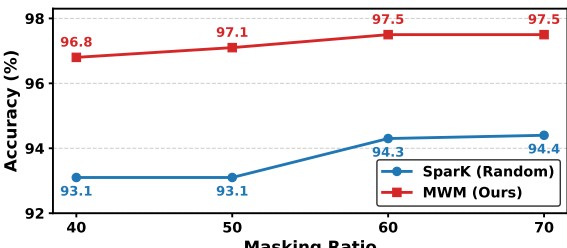

Figure 4: Classification performance on the Chest CT dataset under different masking strategies and ratios.

the robustness of our comparison. These results challenge the prevailing assumption that high masking ratios are inherently optimal for masked image modeling. Instead, they highlight the advantage of semantically guided, task-aware masking strategies like MWM, which achieve stronger downstream performance with lower masking ratios by preserving task-relevant content and reducing redundancy.

## 5 CONCLUSION AND LIMITATIONS

This work presents **Mask What Matters (MWM)**, a self-supervised pretraining framework that integrates text-guided semantic localization with region-aware masking. By leveraging natural language prompts to highlight task-relevant areas and applying differentiated masking, MWM enables semantically aligned representation learning without requiring per-image annotations.

Comprehensive experiments demonstrate the effectiveness and generalizability of MWM: (1) natural language prompts reliably guide semantic region localization, validating the feasibility of text-driven masking; (2) MWM consistently outperforms existing methods (e.g., SparK, AnatoMask) in classification across three imaging modalities, and also yields gains in detection and instance segmentation; (3) MWM maintains strong performance even at lower masking ratios (e.g., 40%), underscoring the benefit of semantic guidance in self-supervised pretraining.

**Limitations.** While promising, our framework has several limitations. First, although text prompts offer flexible guidance, the robustness of MWM to variations in prompt style, vocabulary, or noise has not been systematically examined. Second, the reliance on external textual descriptions introduces a dependency that may weaken generalization in scenarios where reliable prompts are unavailable or inconsistent.

Looking forward, we believe that text-driven, region-aware masking offers a principled path toward more semantically grounded self-supervised learning, and that extending MWM beyond medical imaging may open new opportunities for broader vision–language pretraining.

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

## A USE OF LARGE LANGUAGE MODELS (LLMS)

We used LLMs *only* for language editing and presentation polishing (e.g., grammar, phrasing, and minor stylistic clarity) of text written by the authors. LLMs were *not* used for idea generation, experimental design, data annotation, code implementation, or analysis/interpretation of results. All technical content, algorithms, proofs, figures, tables, metrics, and conclusions were authored and verified by the authors. No confidential or identifying data were provided to LLMs beyond anonymized manuscript text.

