# OpenReview forum: "Mask What Matters: Controllable Text-Guided Masking for Self-Supervised Medical Image Analysis"
_ICLR.cc/2026/Conference — Submitted to ICLR 2026_

### Official Review · Reviewer_2f32 · 2025-10-31

**Soundness:** 3
**Presentation:** 3
**Contribution:** 2
**Rating:** 2
**Confidence:** 4

**Summary:**

This paper presents a method to improve the masked image modeling for medical image self-supervised learning. The method proposes to use Biomedclip to identify the semantically important regions in medical images and then assign higher masking ratios to the relevant regions. Experiments are conducted on three types of medical imaging modalities with comparison to existing masked image modeling methods.

**Strengths:**

- The paper is clearly written and easy to follow.
- The method design is well motivated.
- The proposed masked image modeling method consistently improves the performance on downstream tasks.

**Weaknesses:**

- The main idea of the paper lacks sufficient novelty. The concept of using region localization to guide masking has been explored in some prior studies, such as [R1]. The manuscript does not clearly describe how the proposed method differs from existing approaches. A more comprehensive review of related region-aware or localization-guided self-supervised methods would strengthen the paper.

- It remains unclear how the accuracy of region localization affects the overall effectiveness of the method. As shown in Table 1, the localization module itself demonstrates low precision, which raises concerns about its benefit to the pretraining.

- The proposed method appears to rely on several sensitive hyperparameters, including the region expansion strategy and the masking ratios for tumor/organ regions and background regions. However, the paper does not analyze how these design choices affect the final performance. An ablation or sensitivity study would help clarify their influence and stability.

- The datasets and tasks used for evaluation are relatively limited and may not be sufficient to convincingly demonstrate the method’s effectiveness. The three datasets used are quite small. When converted from slices to volumetric cases, they represent only a few hundred CT and MRI volumes. It would be valuable to test the method on larger datasets to show its effectiveness.

- The experimental comparison is limited. The paper should include comparisons with other medical image-specific self-supervised learning methods, as well as SOTA approaches relevant to each downstream task.

- The linear probing for the classification task and detection and segmentation experiments are only compared with AnatoMask and SparK methods, which are limited. The comparison with other self-supervised learning tasks should be involved.

- For segmentation tasks, standard evaluation metrics such as the Dice coefficient and the Hausdorff Distance should be reported.

Reference:
[R1] Xie Y, Gu L, Harada T, Zhang J, Xia Y, Wu Q. Medim: Boost medical image representation via radiology report-guided masking. In International Conference on Medical Image Computing and Computer-Assisted Intervention, 2023.

**Questions:**

- How does the proposed method differ from previous region-relevant masking approaches in medical image self-supervised learning?

- How would the performance compare if ground-truth labels were used to specify the regions instead of the proposed localization approach? In addition to BiomedCLIP, what other possible alternatives were considered for region specification, and how might these choices affect the overall performance?

- How sensitive is the method to the important hyperparameters?

- How would the method perform on larger datasets and more difficult classification tasks?

- How would the method compare to other medical image self-supervised learning methods and SOTA approaches relevant to each downstream task?

---

### Official Review · Reviewer_G1op · 2025-11-01

**Soundness:** 3
**Presentation:** 3
**Contribution:** 3
**Rating:** 4
**Confidence:** 4

**Summary:**

This paper introduces "Mask What Matters" (MWM), a novel self-supervised pretraining framework for medical image analysis that combines text-guided semantic localization with region-aware masked image modeling. The key innovation is leveraging vision-language models (specifically BiomedCLIP) along with the Multi-Modal Information Bottleneck (M2IB) and Segment Anything Model (SAM) to identify task-relevant regions from open-vocabulary text prompts. These regions then receive differentiated masking ratios during pretraining—higher masking for semantically important areas (e.g., lesions/organs) and lower for background regions. The authors demonstrate consistent improvements over state-of-the-art masked image modeling methods like SparK and AnatoMask across multiple medical imaging modalities (brain MRI, chest CT, lung X-ray) for classification, detection, and segmentation tasks, achieving these gains with substantially lower overall masking ratios (40% vs. 70%).

**Strengths:**

- **Innovative method integration:** The creative fusion of text-guided semantic localization with masked image modeling is a compelling solution to the common issue of semantic misalignment in random masking. This novel approach effectively connects vision-language models with self-supervised learning to better prioritize diagnostically relevant regions.
- **Extensive and diverse evaluation:** The paper rigorously evaluates the method across multiple medical imaging modalities (brain MRI, chest CT, lung X-ray) and a variety of downstream tasks (classification, detection, segmentation), providing strong evidence of broad applicability and robustness.
- **Practical efficiency improvements:** Demonstrating competitive or superior performance using a substantially lower masking ratio (40%) compared to conventional high ratios (70%) represents a significant advancement in computational efficiency and practical deployment.
- **Annotation-free and flexible framework:** Relying solely on general text prompts rather than per-image annotations lowers the barrier for widespread adoption and enables easy integration with different backbone architectures and imaging modalities.
- **Well-designed ablation and analysis:** The inclusion of detailed ablation studies on prompt formulation, SAM-based region refinement, and masking ratio impacts offers valuable insights into the contribution of each component toward overall performance gains.

**Weaknesses:**

- **Incremental technical novelty:** While the integration is thoughtful, the work primarily combines existing components (BiomedCLIP, M2IB saliency estimation, SAM refinement) without proposing fundamentally new algorithms or theoretical breakthroughs in masked image modeling.
- **Moderate improvement margins:** The performance gains, mostly in the range of 1-3%, though consistent and meaningful, may be perceived as modest relative to the added architectural and computational complexity of the multi-stage masking pipeline.
- **Lack of computational cost assessment:** The paper does not provide an analysis of the computational overhead introduced by the additional text-guided localization steps, which involve multiple external models and could impact efficiency in resource-constrained settings.
- **Limited scalability and generalization testing:** Evaluations are carried out on relatively modest-sized medical datasets. The framework's scalability to much larger datasets or adaptation to natural image domains remains an open question.
- **Insufficient failure case analysis:** There is little discussion or empirical evaluation of scenarios where the text-guided localization yields inaccurate or suboptimal regions, nor how such failures affect downstream learning and performance robustness.
- **Dependence on pretrained models:** The approach’s reliance on pretrained vision-language (BiomedCLIP) and segmentation (SAM) models introduces a dependency that may affect robustness, domain adaptability, and future applicability if model quality or availability changes.
- **Limited to 2D medical images:** The method has been developed and validated primarily on 2D imaging modalities due to the current availability of pretrained vision-language and segmentation models. This limits applicability to real-world clinical use cases involving 3D volumetric medical scans, where equivalent models and tools are not yet available, posing challenges for practical deployment in comprehensive 3D medical image analysis.

**Questions:**

1. **Clarification on Computational Overhead:**
   Could you provide a detailed analysis or estimation of the computational costs introduced by the text-guided localization pipeline, including the use of BiomedCLIP, M2IB, and SAM? Understanding the runtime and hardware requirements is critical for assessing practical applicability.

2. **Scalability and Dataset Size:**
   Given the relatively modest dataset sizes (17K-39K images) used for evaluation, how well do you expect your method to scale to larger, more diverse datasets or to natural image domains? Are there plans or preliminary results for scaling?

3. **Generalization to 3D Medical Imaging:**
   Since the current evaluation is on 2D imaging modalities due to pretrained model availability, could you elaborate on the challenges and potential approaches for extending this framework to 3D volumetric medical images? Do you foresee progress in 3D pretrained models enabling this soon?

4. **Failure Modes and Robustness:**
   Can you provide any analysis or examples where the text-guided localization fails or produces suboptimal regions? How sensitive is the downstream learning to such imperfect region localizations, and is there a mechanism to mitigate adverse effects?

5. **Prompt Design Sensitivity:**
   The ablation studied phrase versus descriptive sentence prompts. Could you provide more guidance on the prompt creation process? For example, what level of domain expertise is required, and how robust is performance to variations or noise in prompt wording?

6. **Dependency on External Models:**
   As BiomedCLIP and SAM models are central to your approach, how do you handle possible domain shift or biases in these pretrained models? Would fine-tuning or adaptation to specific tasks or modalities improve the results?

7. **Practical Deployment Considerations:**
   From your perspective, what are the main obstacles to deploying this method in clinical workflows today? Beyond model availability, are there regulatory, interpretability, or user-integration hurdles you anticipate?

8. **Potential for Simplified or End-to-End Solutions:**
   The current pipeline is multi-stage and involves different specialized modules. Do you see a path toward more streamlined or end-to-end trainable architectures that integrate semantic guidance without substantial modular complexity?

---

### Official Review · Reviewer_2yks · 2025-11-03

**Soundness:** 2
**Presentation:** 2
**Contribution:** 2
**Rating:** 2
**Confidence:** 5

**Summary:**

This paper introduces Mask What Matters, a new framework for self-supervised medical image analysis using controllable text-guided masking. By integrating visual language models such as BiomedCLIP, Multi-Modal Information Bottleneck (M2IB), and Segment Anything Model (SAM) for cue-driven region localization, the method adaptively applies a differential masking strategy to prioritize diagnostically salient regions while reducing redundancy in non-informative background regions.

**Strengths:**

1.The paper introduces a controllable text-guided masking mechanism for the first time, using visual-language models (such as BiomedCLIP and SAM) to achieve zero-shot region localization, avoiding the semantic mismatch problem of traditional random masks. This allows the mask to focus more on diagnostically relevant areas (such as tumors) rather than redundant background.
2.The framework is unsupervised, requires no labeled data, and is backbone-agnostic (supporting both ViT and ConvNet). Through multi-level masking and hierarchical reconstruction, the mask ratio is reduced (from 70% to 40%) while maintaining high reconstruction quality.

**Weaknesses:**

1. This paper lacks sufficient description and analysis of text processing. Moreover, the use of both “Phrase” and “Sentence” prompts in the experiments lacks adequate justification, and directly validating these two types of prompts on downstream tasks appears methodologically questionable.It is hoped that the authors can use experiments with Prec. and Recall to demonstrate the difference between these two types of prompts.
2. The paper aims to replace traditional random masking with a text-guided masking strategy (MWM) to reduce occlusion of irrelevant regions. However, the relatively low Precision reported in Table 1 suggests that MWM may not effectively address this issue. It would be beneficial for the authors to include a comparison with existing methods to further validate the effectiveness of their approach.
3. The paper points out that FocusMAE suffers from limited generalization ability. However, the proposed method also relies on using different prompts in the LLM for text processing, which may introduce similar generalization issues.
4.The experimental results in Table 5 are somewhat confusing.

**Questions:**

1. This paper lacks sufficient description and analysis of text processing. Moreover, the use of both “Phrase” and “Sentence” prompts in the experiments lacks adequate justification, and directly validating these two types of prompts on downstream tasks appears methodologically questionable.It is hoped that the authors can use experiments with Prec. and Recall to demonstrate the difference between these two types of prompts.
2. The paper aims to replace traditional random masking with a text-guided masking strategy (MWM) to reduce occlusion of irrelevant regions. However, the relatively low Precision reported in Table 1 suggests that MWM may not effectively address this issue. It would be beneficial for the authors to include a comparison with existing methods to further validate the effectiveness of their approach.
3. The paper points out that FocusMAE has limited generalization ability. However, the proposed method also relies on different prompts in LLM for text processing, which may introduce similar generalization problems. The paper would be more convincing if the authors could provide experimental evidence or further discussion clarifying how MWM mitigates this potential limitation.
4. I have several concerns regarding the results presented in Table 5, and I would appreciate clarification from the authors.
(1) The conclusions drawn from the choice of “Phrase” and “Sentence” prompts on the Lung X-ray dataset differ from those obtained on the other two datasets. What are the underlying differences between the chest X-ray dataset and the others that might explain this inconsistency?
(2) In the experiment analyzing the Impact of SAM, which type of prompt (“Phrase” or “Sentence”) was used? Additionally, why do the results of w/o SAM differ from those reported in the "Prompt Type" experiments?

---

### Meta-Review · Area_Chair_6v9d · 2026-01-01

**Summary:**

In specialized fields such as medical imaging, there is a severe shortage of annotated data. Although self-supervised MIM shows promise, existing methods rely on high-ratio random masking, which results in inefficiency and low semantic consistency. Additionally, domain-aware approaches depend on heuristics or supervised signals, which limits their adaptability. Therefore, this paper proposes "Mask What Matters," a controllable, text-guided masking method that emphasizes important regions and reduces background redundancy through prompt-based localization using a vision-language model (VLM).

 Reviewers raised several concerns. For example: "The explanation of text processing is insufficient. The rationale and differences for combining 'phrase/sentence' prompts should be demonstrated experimentally"; "While MWM aims to suppress irrelevant regions, its low precision raises doubts about its effectiveness"; and "Concerns exist that prompt dependency compromises generalizability. The inconsistencies in Table 5 and the consistency between SAM/prompt conditions should be explained."

The authors did not submit rebuttals addressing these concerns, leaving them unresolved. Therefore, the AC has decided to reject this paper.

**Reviewer Concerns:**

Because the authors did not submit a rebuttal, all concerns remain unresolved.

**Reviewer Scores:**

This paper received negative evaluations from all reviewers. Since the authors did not submit a rebuttal addressing the reviewers’ concerns, the scores did not change.

---

### Decision · Program_Chairs · 2026-01-26

Reject